# TRIM21 Promotes Oxidative Stress and Ferroptosis through the SQSTM1-NRF2-KEAP1 Axis to Increase the Titers of H5N1 Highly Pathogenic Avian Influenza Virus

**DOI:** 10.3390/ijms25063315

**Published:** 2024-03-14

**Authors:** Yifan Wei, Yongxia Gu, Ziwei Zhou, Changrong Wu, Yanwei Liu, Hailiang Sun

**Affiliations:** 1College of Veterinary Medicine, South China Agricultural University, Guangzhou 510642, China; weiyifan202401@163.com (Y.W.); gyx1044510376@163.com (Y.G.); 15716486758@163.com (Z.Z.); wuchangrong0809@163.com (C.W.); liuyanwei2412@163.com (Y.L.); 2Key Laboratory of Zoonosis Control and Prevention of Guangdong Province, South China Agricultural University, Guangzhou 510642, China; 3National and Regional Joint Engineering Laboratory for Medicament of Zoonosis Prevention and Control, South China Agricultural University, Guangzhou 510642, China

**Keywords:** TRIM21, ferroptosis, oxidative stress, influenza virus, NRF2

## Abstract

Tripartite motif-containing protein 21 (TRIM21) is involved in signal transduction and antiviral responses through the ubiquitination of protein targets. TRIM21 was reported to be related to the imbalance of host cell homeostasis caused by viral infection. Our studies indicated that H5N1 highly pathogenic avian influenza virus (HPAIV) infection up-regulated TRIM21 expression in A549 cells. Western blot and qPCR results showed that knockdown of TRIM21 alleviated oxidative stress and ferroptosis induced by H5N1 HPAIV and promoted the activation of antioxidant pathways. Co-IP results showed that TRIM21 promoted oxidative stress and ferroptosis by regulating the SQSTM1-NRF2-KEAP1 axis by increasing SQSTM1 K63-linked polyubiquitination under the condition of HPAIV infection. In addition, TRIM21 attenuated the inhibitory effect of antioxidant NAC on HPAIV titers and enhanced the promoting effect of ferroptosis agonist Erastin on HPAIV titers. Our findings provide new insight into the role of TRIM21 in oxidative stress and ferroptosis induced by viral infection.

## 1. Introduction

In the last century, avian influenza outbreaks were brought about by various H5 subtype viruses in twenty-three nations and areas [1]. Epidemiological investigations have shown that the route of infection in humans is related to contact with virus-infected poultry or a polluted environment [2]. There have been 878 recorded human cases, with a fatality rate of more than 50% (https://www.who.int/publications, accessed on 17 January 2024). The circulation of avian influenza viruses causes significant economic losses in the poultry industry and poses a serious threat to public health [3,4]. However, the pathogenicity mechanism of AIV is still not clear.

Hosts have developed various non-enzymatic and enzymatic antioxidants to detoxify reactive oxygen species (ROS) and alleviate oxidative stress, including glutathione, thioredoxin, superoxide dismutase (SOD), catalase, and peroxidase [5,6,7,8]. The NRF2/ARE signaling pathway is the main antioxidant pathway [9,10]. NRF2 is a highly sensitive transcription factor that regulates the antioxidant capacity of cells [11,12]. Some viruses regulate oxidative stress by regulating the NRF2 signal pathway [13,14,15]. The NS2B3 complex of DENV degraded the NRF2 to inhibit the antioxidant gene network, resulting in increased ROS levels and promoting viral replication [16]. NRF2 is regulated by the SQSTM1-NRF2-KEAP1 axis [17]. Influenza virus infection can also cause oxidative stress [18].

Oxidative stress can cause ferroptosis [19,20]. This is a new form of cell death, characterized mainly by the accumulation of ROS, ferrous ions, and phospholipid peroxides [21,22,23,24,25]. Glutathione, an antioxidant tripeptide composed of glutamic acid, cysteine, and glycine [5], is an important regulator of cellular antioxidant defense. The consumption of cystine or glutathione leads to the accumulation of lipid ROS and the promotion of ferroptosis [26,27]. A variety of viruses promote their replication by regulating ferroptosis [28,29]. A virus can induce ferroptosis by regulating host cell iron metabolism, lipid metabolism, and antioxidant systems [30]. The Newcastle disease virus induces ferroptosis to promote replication by up-regulating the p53 pathway to inhibit the Xc-system and cause iron autophagy [31]. The influenza virus can also cause ferroptosis [32].

TRIM proteins are a multifunctional family of E3 ubiquitin ligases involved in a number of biological activities [33]. TRIM family proteins play crucial roles in the regulation of gene transcription, antiviral response, apoptosis, innate immunity, and development [34,35,36]. TRIM21 regulates a series of viral infections by affecting the innate immunity pathway [37]. Currently, TRIM21 is considered to regulate the antioxidant pathway and induce ferroptosis [38,39,40].

However, the role of TRIM21 in modulating H5N1 HPAIV-induced oxidative stress and ferroptosis is still not clear. The purposes of the present study are (1) to detect oxidative stress and ferroptosis induced by the H5N1 influenza virus in A549 cells; (2) to study the effects of TRIM21 on the antioxidant pathway and virus-induced ferroptosis; and (3) to demonstrate the role of TRIM21 in regulating the SQSTM1-NRF2-KEAP1 axis.

## 2. Results

### 2.1. H5N1 Avian Influenza Virus Infection Induces Oxidative Stress and Ferroptosis

A variety of viral infections can cause oxidative stress in cells. We used DK/212 to infect A549 cells and detected the intracellular ROS level at 12, 24, and 36 h. Compared to the control group, the ROS level of DK/212-infected cells increased at 24 h and 36 h (Figure 1A). We also detected the changes in mitochondrial membrane potential at different time points after virus infection, and the Carbonyl cyanide 3-chlorophenylhydrazone (CCCP)-treated cell group was used as a positive control. The results showed that DK212 infection caused a decrease in mitochondrial membrane potential (Figure 1B). Oxidative stress and mitochondrial membrane potential decrease are important factors in ferroptosis, so we detected whether H5N1 HPAIV infection could induce ferroptosis in cells. First, we detected the intracellular ferrous iron level after DK/212 infection. The results showed that the intracellular ferrous iron level increased significantly (*p* < 0.001), which was similar to the effect of ferroptosis inducer Erastin (*p* < 0.01) (Figure 1C). Then, we detected the level of lipid peroxide marker-malondialdehyde (MDA) in the infected group. The experimental results showed that the level of MDA in the infected group was significantly increased (*p* < 0.0001), but the accumulation of MDA in the cells was inhibited after treatment with the ferroptosis inhibitor Lip-1 (*p* < 0.01) (Figure 1D). Finally, we detected the level of intracellular total glutathione (GSH) and cell viability. The results showed that the level of intracellular GSH was lower than that of the control group after DK/212 infection (*p* < 0.001) (Figure 1E), and the cell viability was also significantly reduced (*p* < 0.01). After treatment with Lip-1, the level of GSH in the infected group increased (*p* < 0.05), and the cell viability increased (*p* < 0.05) (Figure 1F). In summary, H5N1 HPAIV infection induces oxidative stress and ferroptosis in A549 cells.

### 2.2. H5N1 AIV Infection Up-Regulated the Expression of TRIM21 and Inhibited Antioxidant Genes

We detected the expression of TRIM21 after DK/212 infection in A549 cells. The results showed that the expression of TRIM21 mRNA was significantly increased (*p* < 0.001) and protein levels also increased (*p* < 0.05) (Figure 2A–C). The mRNA transcription level of antioxidant downstream genes Solute Carrier Family 7 Member 11 (SLC7A11) decreased (*p* < 0.05) after DK/212 infection. The mRNA transcription level of antioxidant downstream genes Glutathione Peroxidase 4 (GPX4) decreased (*p* < 0.01) after DK/212 infection. The mRNA transcription level of antioxidant downstream genes heme oxygenase 1 (HO-1) decreased (*p* < 0.05) after DK/212 infection. The mRNA transcription level of antioxidant downstream genes Quinone oxidoreductase 1 (NQO1) decreased (*p* < 0.05) after DK/212 infection. The mRNA levels of glutamate-cysteine ligase GCLC and GCLM were also down-regulated (*p* < 0.01, *p* < 0.001) (Figure 2D–J) The primers of related genes are shown in Table 1. In addition, we found that the protein expression levels of NRF2, SLC7A11, and GPX4 were down-regulated after viral infection (Figure 2K). These results indicate that H5N1 HPAIV infection promotes the expression of TRIM21 and down-regulates the expression of antioxidant pathway genes in A549 cells.

### 2.3. Interference with TRIM21 Alleviated the Accumulation of Ferroptosis Markers and Oxidative Stress Caused by H5N1 AIV

To confirm whether TRIM21 induces oxidative stress in DK/212-infected A549 cells, we designed three primers of Si TRIM21 for interference (Table 2). We found that Si RNA2 showed a good interference effect (Figure 3A–C). After knocking down TRIM21 in A549 cells, we used DK/212 to infect A549 cells. Compared to the control group, the increase in ROS caused by DK/212 infection after interference with TRIM21 was significantly reduced at 24 and 36 h (Figure 3D). To confirm whether TRIM21 could alleviate ferroptosis caused by H5N1 HPAIV infection, we detected the level of MDA and GSH in Si TRIM21 A549 cells infected with DK/212. The results showed that the Si-TRIM21 group significantly inhibited the accumulation of MDA in cells after virus infection (Figure 3E). The Si TRIM21 group also significantly increased the total glutathione level in the cells, and there was no difference between the LIP-1-treated infection groups (Figure 3F). Interference with TRIM21 did not affect cell viability (Figure 3G).

### 2.4. TRIM21 Affects Cell Antioxidant Function by Down-Regulating NRF2

To determine whether TRIM21 can affect the antioxidant pathway to promote ferroptosis, we knocked down the expression of the TRIM21 gene and infected A549 cells with DK/212 to detect the expression of related genes. The results showed that, compared to the control group, the down-regulation of antioxidant gene mRNA caused by DK/212 infection was inhibited after interference with the expression of the TRIM21 gene (Figure 4A–F), and the protein level of NRF2, SLC7A11, and GPX4 was increased (Figure 4G). Next, we detected whether TRIM21 affected the antioxidant pathway by affecting the NRF2. NRF2 was overexpressed in A549 cells and then infected with DK/212. Compared to the control group, the protein levels of SLC7A11 and GPX4 proteins were increased in A549 cells (Figure 4H). Next, cells were treated with NRF2 specific inhibitor ML385 to inhibit NRF2 function as the NRF2 inhibitor group, with Si-TRIM21 to inhibit TRIM21 expression as the TRIM21 interference group, with ML385 and Si-TRIM21 as the NRF2 and TRIM21 co-interference group, and with DMSO and Si-control as the control group. These groups were infected with DK/212. The results showed that the SLC7A11 and GPX4 protein levels in the NRF2 and TRIM21 co-interference group were similar to those of the control group and lower than those of the TRIM21 interference group (Figure 4I). This indicated that the overexpression of NRF2 increased the total glutathione level in the cells and reduced the level of malondialdehyde, which was similar to the results of the interference TRIM21 group (Figure 4J,K). In summary, TRIM21 affects the antioxidant pathway by regulating NRF2.

### 2.5. TRIM21 Competitively Binds to SQSTM1 to Regulate KEAP1/NRF2 Axis

The NRF2 is mainly affected by the KEAP1-NRF2 axis. Our previous studies found that H5N1 HPAIV affects the SQSTM1-NRF2-KEAP1 axis by down-regulating TRIM16 to disrupt the intracellular antioxidant system [41]. Here, we detected whether TRIM21 plays a similar function to TRIM16. HEK293T cells were co-transfected with TRIM21-HA and SQSTM1-FLAG, TRIM21-HA and NRF2-FLAG, and TRIM21-HA and KEAP1-FLAG plasmids for co-immunoprecipitation, respectively. The results showed that TRIM21 did not directly interact with KEAP1 and NRF2 but could interact with SQSTM1/P62 (Figure 5A–C). In addition, we co-transfected eGFP-TRIM21 and AsREDs-SQSTM1 plasmids in HEK 293T cells and observed the co-localization of TRIM21 and SQSTM1 proteins in the cytoplasm by laser confocal experiments (Figure 5D). SQSTM1 can chelate KEAP1 to release NRF from the KEAP1-NRF2 complex. Therefore, we co-expressed TRIM21 with or without SQSTM1-FLAG and KEAP1-HA in HEK293T cells and observed that the interaction between SQSTM1 and KEAP1 was reduced in the case of the overexpression of TRIM21-eGFP. This indicated that TRIM21 and KEAP1 competed to bind to SQSTM1 (Figure 5E). We co-expressed TRIM21-HA and SQSTM1-FLAG in A549 cells and observed that DK/212 infection enhanced the interaction between TRIM21 and SQSTM1 (Figure 5F). In combining this with previous results, we believe that DK/212 infection enhances the expression of TRIM21 and competitively binds to SQSTM1 to reduce the release of NRF2 from the KEAP1-NRF2 complex, which ultimately destroys the antioxidant system of cells and promotes ferroptosis.

### 2.6. TRIM21 Modifies SQSTM1 Protein by K63 Chain Ubiquitination 

To detect whether SQSTM1 is ubiquitinated by TRIM21 in the normal state of cells, we constructed the UB-HA plasmid and co-transfected with TRIM21-eGFP and SQSTM1-FLAG into 293T cells. The empty plasmid vector, SQSTM1-FLAG, and UB-HA were co-transfected as a control for immunoprecipitation experiments. The results showed that the overexpression of TRIM21 promoted the ubiquitination of SQSTM1 protein (Figure 6A). To explore how TRIM21 affected SQSTM1, we constructed two plasmids, UB (K48)-HA and UB (K63)-HA. The Co-IP results showed that the polyubiquitination of K48-linked NRF2 was similar to that of ordinary polyubiquitination, but TRIM21 protein promoted an increase in the polyubiquitination of K63-linked SQSMT1 (Figure 6B). In summary, TRIM21 affects SQSTM1 ubiquitination by increasing k63-linked polyubiquitination.

### 2.7. TRIM21 Affects Viral Titer by Promoting Oxidative Stress and Ferroptosis

We used NAC or Erastin to treat A549 cells infected with H5N1 HPAIV, and we detected the viral titers at 24 and 36 h (Figure 7A). NAC reduced the virus titers, but Erastin increased the virus titers. We overexpressed TRIM21 in A549 cells and detected the virus titers after treatment with NAC or Erastin. The results showed that the overexpression of TRIM21 increased the virus titers in the NAC group and increased the virus titer only at 36 h in the Erastin group (Figure 7B,C). In summary, TRIM21 can affect the titers of AIV by promoting oxidative stress and ferroptosis.

## 3. Discussion

Reactive oxygen species (ROS) and antioxidant ingredients are crucial signaling molecules in oxidative stress response [42]. HCV, SARS-CoV-2, and DENV infection can induce oxidative stress and the accumulation of ROS [43,44]. The PB2, NS1, and M2 proteins of the influenza virus can regulate mitophagy to induce the accumulation of ROS to promote viral replication [45,46,47]. H5N1 virus infection can inhibit the expression of SOD1 (Superoxide Dismutase 1) and promote the accumulation of ROS in cells [48]. The NRF2/KEAP1 pathway plays an important role in eliminating ROS. Under stress conditions, NRF2 dissociates from the NRF2/KEAP1 complex and enters the nucleus to interact with the antioxidant response element (ARE) to up-regulate the expression of antioxidant genes. The NRF2 antioxidant network was activated to protect cells from damage by influenza virus infection [49]. The phosphorylated form of NRF2 is not imported to the nucleus at the early stage of H5N1 and H7N9 virus infection, indicating that a highly pathogenic influenza virus inhibits the activation of the NRF2 pathway [50]. Knockdown of PSMA2 inhibited the nuclear translocation of NRF2, leading to the accumulation of ROS and affecting the maturation of the influenza virus in A549 cells [51]. The influenza virus affects the activity of G6PD to promote the occurrence of oxidative stress by inhibiting the expression of SIRT2 and NRF2 in cells [52]. Knockout NRF2 increased the susceptibility of alveolar epithelial cells to influenza A virus infection, while the overexpression of NRF2 reduced viral replication [53]. Antioxidant n-acetylcysteine (NAC) can inhibit the replication of the H5N1 influenza virus in A549 cells [54]. Our findings demonstrated that H5N1 HPAIV infection promoted the accumulation of ROS and decreased mitochondrial membrane potential in A549 cells, which was consistent with the previous study [47]. Western blot and qPCR experiments showed that oxidative stress induced by virus infection was related to the down-regulation of NRF2 expression and the inhibition of antioxidant pathways.

Ferroptosis is one type of regulatory cell death, which is an important strategy for host cells to resist pathogen invasion. RSL3 is a ferroptosis inducer that has been reported to inhibit the proliferation of porcine epidemic diarrhea virus (PEDV) in Vero cells [55]. Brequinar activates ferroptosis to inhibit ASFV replication [56]. However, a variety of viruses can use host ferroptosis to promote their replication, transmission, and pathogenicity [28]. EB virus up-regulates SLC7A11 and GPX4 protein levels by affecting the SQSTM1-NRF2-KEAP1 axis to promote the formation of GPX4 and TBK1 complexes and to regulate downstream pathways to reduce the sensitivity of NPC cells to ferroptosis [57]. Herpes simplex virus-1 (HSV-1) promotes KEAP1-dependent ubiquitination and degradation of NRF2 to induce ferroptosis and inflammation in human astrocytes (U373) and microglia (HMC3) cells [58]. Newcastle disease virus [31], H3N2 swine influenza virus, and H1N1 influenza virus infection [59,60] can cause ferroptosis, which is characterized mainly by the accumulation of ferrous ions, ROS, and MDA and a decreased GSH level. Here, we found that H5N1 HPAIV infection increased the levels of ROS, MDA, and ferrous ions and reduced mitochondrial membrane potential and GSH levels. Ferroptosis inhibitor treatment recovered some cell activity after H5N1 HPAIV infection. These findings indicated that H5N1 HPAIV can induce ferroptosis in A549 cells. SLC7A11 and GPX4 are important proteins in the ferroptosis pathway [61]. We found that H5N1 HPAIV down-regulated the transcription and expression of SLC7A11 and GPX4. These results are consistent with those of NDV, H3N2 influenza virus, and H1N1 influenza virus infection. Knockdown of KEAP1 in nasal epithelial cells increased the expression of NRF2 and enhanced the transcription of SLC7A11 and GPX4 to alleviate ferroptosis induced by H1N1 influenza virus infection [56]. NRF2 plays an important role in mitigating lipid peroxidation and ferroptosis [62]. In the present study, knockdown of TRIM21 increased the expression of NRF2, SLC7A11, and GPX4 in A549 cells and alleviated ferroptosis induced by H5N1 HPAIV infection. The overexpression of NRF2 enhanced the expression of SLC7A11 and GPX4 in A549 cells and alleviated ferroptosis induced by virus infection. Further studies showed that simultaneous inhibition of TRIM21 and NRF2 did not increase the expression of SLC7A11 and GPX4 in A549 cells. These results indicated that knockdown of TRIM21 alleviated ferroptosis by increasing the expressions of NRF2, SLC7A11, and GPX4 in A549 cells infected with H5N1 HPAIV; these findings are consistent with those in the EBV study [56]. 

TRIM21 can regulate innate immune signaling pathways and virus replication [37]. TRIM21 ubiquitination degrades IRF7 to promote rabies virus replication [63]. NMI enhances H1N1 influenza virus replication by promoting TRIM21 to degrade IRF7 [64]. TRIM21 regulates the SQSTM1-NRF2-KEAP1 axis to regulate oxidative stress homeostasis [40]. The NS protein of severe fever with thrombocytopenia syndrome virus (SFTSV) regulates the SQSTM1-NRF2-KEAP1 axis to stimulate the activation of antioxidant pathways and promote virus replication by interacting with TRIM21 [65]. H5N1 HPAIV infection down-regulated TRIM16 in A549 cells and the overexpression of TRIM16 inhibited H5N1 HPAIV infection-induced ROS accumulation and reduced virus titers. TRIM16 regulated the SQSTM1-NRF2-KEAP1 axis and increased NRF2 expression by increasing NRF2 K63-linked polyubiquitination to stimulate the activation of the antioxidant pathway [41]. Here, we found that H5N1 HPAIV infection increased TRIM21 in A549 cells. TRIM21 increased SQSTM1 K63-linked polyubiquitination to negatively regulate the SQSTM1-NRF2-KEAP1 axis to inhibit the activation of the antioxidant pathway. These results indicate that H5N1 HPAIV infection regulates the SQSTM1-NRF-KEAP1 axis to regulate antioxidant pathways through different TRIM proteins. TRIM21 ubiquitination degrades GPX4 to promote ferroptosis in acute kidney injury [39]. Knockout of TRIM21 in MEF and H9c2 cells enhanced p62 sequestration of KEAP1 and inhibited ferroptosis induced by doxorubicin [66]. The regulation mechanism of TRIM21 in virus-induced ferroptosis is still unclear. We found that knockdown of TRIM21 stimulated the activation of antioxidant pathways, reduced the accumulation of ROS and MDA, and increased the level of intracellular GSH. These results indicate that TRIM21 promotes ferroptosis induced by H5N1 HPAIV infection, but the molecular mechanism of TRIM21 in regulating ferroptosis requires further study.

## 4. Materials and Methods

### 4.1. Viruses and Cells

The H5N1 HPAIV, A/duck/Guangdong/212/2004 (DK/212) [39] was isolated and stored in the laboratory. The virus was propagated in 9–11 days in specific pathogen-free chicken embryos. Human lung carcinoma cells (A549) and human embryo kidney cells (HEK293T) were grown in Dulbecco’s modified Eagle’s medium (DMEM, GIBCO, Grand Island, NE, USA) supplemented with 10% fetal bovine serum (FBS, GIBCO, USA), 100 U/mL penicillin, and 100 mg/mL streptomycin. The cells were incubated at 37 °C, 5% (*v*/*v*) CO_2_. All experiments involving H5N1 HPAIV were conducted in the biosafety level 3 laboratory (BSL-3). A549 cells were infected with DK/212 at an MOI of 0.1 at 37 °C. Following a one-hour absorption period, unattached viruses were removed. The cells were then washed with preheated phosphate-buffered saline (PBS) three times and cultured in maintenance medium at 37 °C. The supernatant was collected at 24 and 36 h post infection (hpi) and titrated in MDCK cells.

### 4.2. Construction of Plasmids

The human TRIM21 gene (NM_003141.4) was obtained from the cDNA of A549 cells by polymerase chain reaction (PCR). The purified PCR product was then cloned into the pCAGGS vector, which contains an HA tag in the N terminal, a Flag tag in the C terminal, and an eGFP or AsRed tag in the N terminal. The coding sequences of human SQSTM1 (NM_003900.5), NRF2 (NM_006164.5), and KEAP1 (NM_203500.2) from A549 cells with Flag, HA, and eGFP tags were cloned into the pCAGGS vector to obtain pCAGGS-HA-SQSTM1, pCAGGS-SQSTM1-Flag, pCAGGS-eGFP-Nrf2, pCAGGS-Flag-NRF2, and pCAGGS-HA-KEAP1 expression recombinant plasmids. All recombinant plasmids were identified by sequencing.

The HEK293T cells were plated into a 6-well plate, and plasmids were transfected into the cells with 80–90% confluent. An amount of 4 μg plasmids and 4 μL lipo2000 were mixed in 500 μL Opti-MEM and stood for 15~20 min. HEK293T cells were washed with PBS three times, and added to the mixture. After transfection for 3~4 h, the original medium was discarded, and the cells were washed with PBS twice, and then cultured in 2 mL medium containing 2% FBS.

### 4.3. ROS Detection

A549 cells infected with DK/212 and control cells were washed twice with PBS at 12, 24, and 36 h. An amount of 5 µM CM-H2DCFA diluted in DMEM medium without FBS (Invitrogen, Waltham, MA, USA) was added and incubated at 37 °C for 30 min. Then, the cells were washed with PBS twice and separated by 0.25% trypsin. FACS analysis was performed using the BD FACSCalibur system. At least 10,000 cells were analyzed, and the average channel fluorescence intensity was calculated.

### 4.4. Real-Time Quantitative PCR (RT-qPCR)

The FastPure^○^RCell/Tissue Total RNA Isolation Kit V2 (Vazyme, Nanjin, China) was used to extract total RNA from differently treated cells according to standard instructions. The quality and quantity of RNA in the samples were determined by the uitro-spec2000 mass spectrophotometer. An amount of 1 μg RNA was reverse-transcribed into cDNA using M-ML-V reverse transcriptase (Promega, Fitchburg, WI, USA), Oligo-dT (TaKaRa Bio, Beijing, China), and random primers (TaKaRa). Next, cDNA was used as a template for qPCR detection using Go Taq qPCR Master Mix (Promega, Madison, WI, USA) and SYBR Green I. The qPCR reaction was performed on the CFX96™ real-time system (Bio-Rad, Hercules, CA, USA) under the following cycle conditions: 1 cycle at 95 °C for 2 min, 40 cycles at 95 °C for 15 s, 60 °C for 40 s. Primers for RT-qPCR were designed using the Oligo 7 primer analysis software (Molecular Biology Insights Inc., Cascade, CO, USA) (Table 1). The relative transcription levels of all genes were calculated by 2^−ΔΔCt^, and the GAPDH gene was used as the standard. Each datum represents the results of three independent experiments.

### 4.5. Co-Immunoprecipitation

HEK293T cells were cultured to 80% confluence and then transfected with the associated plasmids with lipo2000 transfection reagent (Invitrogen, Carlsbad, CA, USA). At 24 h post transfection (phi), cells were washed twice with PBS; cell lysate (Beyotime, Nanjin, China) containing 1 nM protease inhibitor (PMSF, Dingguo, Nanjin, China) was added to the cells and incubated at 4 °C for 30 min and then centrifuged at 4 °C for 10 min at 12,000 rpm. The supernatant was incubated with 50 μL agarose beads (Biolinkedin, L-1303, Shanghai, China) and shaken at 4 °C for 6 h. After incubation, agarose beads were washed three times with TBST and boiled in 5×SDS loading buffer (Dingguo, Nanjin, China) for 10 min.

### 4.6. Western Blotting

Cell lysate (Beyotime, Nanjin, China) with 1nM protease inhibitor (PMSF, Dingguo, Nanjin, China) was used to prepare cell lysate products, which were lysed on ice for 30 min and then centrifuged for 10 min at 12,000 rpm at 4 °C. The protein concentration was determined with a BCA kit (Invitrogen, Carlsbad, CA, USA). An amount of 5×SDS loading buffer was added to the sample, which was then boiled for 10 min. Protein lysates were separated on SDS–polyacrylamide gel, transferred onto a nitrocellulose membrane (Millipore, Burlington, MA, USA), and blocked in 5% BSA (Dingguo, Nanjin, China) solution for 1 h. Then, the nitrocellulose membrane was washed with TBST three times and incubated in the primary antibody at 4 °C overnight. The secondary antibodies were DyLight 800 goat anti-mouse or anti-rabbit IgG (H+L) (Thermo Fisher Scientific, Waltham, MA, USA). Primary antibodies used in Western blotting with dilutions were as follows: TRIM21 (Santa Cruz #SC-25351, 1:1000), SQSTM1 (BD #610832, 1:2000), NRF2 (Santa Cruz #SC-365949, 1:1000), KEAP1 (CST #8047S, 1:1000), HA (Thermo Fisher Scientific 20182, 1:2000), Flag (Thermo Fisher Scientific A36803, 1:2000), eGFP (Thermo Fisher Scientific F56-6A1.2.3, 1:2000), SLC7A11 (Abclonal #A13685, 1:1000), and GPX4 (Abclonal #A1933, 1:1000).

### 4.7. Confocal Microscopy

To study the co-localization of TRIM21 and SQSTM1, HEK293T cells were cultured in glass-bottom petri dishes and co-transfected with green fluorescence pCAGGS-eGFP-TRIM21 and red fluorescence pCAGGS-AsRed-SQSTM1 using lipo2000 transfection (Thermo Fisher Scientific, Waltham, MA, USA) reagent. After the co-transfection of plasmids for 24 h, the cells were washed with PBS twice, fixed with 4% paraformaldehyde for 10 min, and infiltrated with 0.1% Triton X-100 at 4 °C for 10 min. After being washed with PBS three times, the cells were stained with 1 µM of 4′,6-diaminyle-2-phenylindole (DAPI) for 5 min. Fluorescence images were obtained using a confocal TCS SP8 microscope (Lecia, Wetzlar, Germany) under a ×100 oil objective lens.

### 4.8. Chemical Reagents

JC-1 (C2006), a GSH and GSSG Assay Kit (S0053), and a Lipid Peroxidation MDA Assay Kit (S0131S) were purchased from Beyotime. Erastin (HY-15763), Liproxstatin-1 (HY-12726), and ML385 (HY-100523) were purchased from MCE (Monmouth Junction, NJ, USA). A Cell Counting Kit-8 was purchased from YRSEN (Nanjin, China). A Ferrous Ion Content Assay Kit (BC5415) was purchased from Solarbio (Beijin, China).

### 4.9. Statistical Analysis

Data were expressed as mean ± standard deviation and statistically analyzed using GraphPad Prism 9 Software (GraphPad Software Inc., San Diego, CA, USA). A Student’s *t*-test was used to evaluate the differences among the groups. The significant differences were set as *p*-value < 0.05.

## Figures and Tables

**Figure 1 ijms-25-03315-f001:**
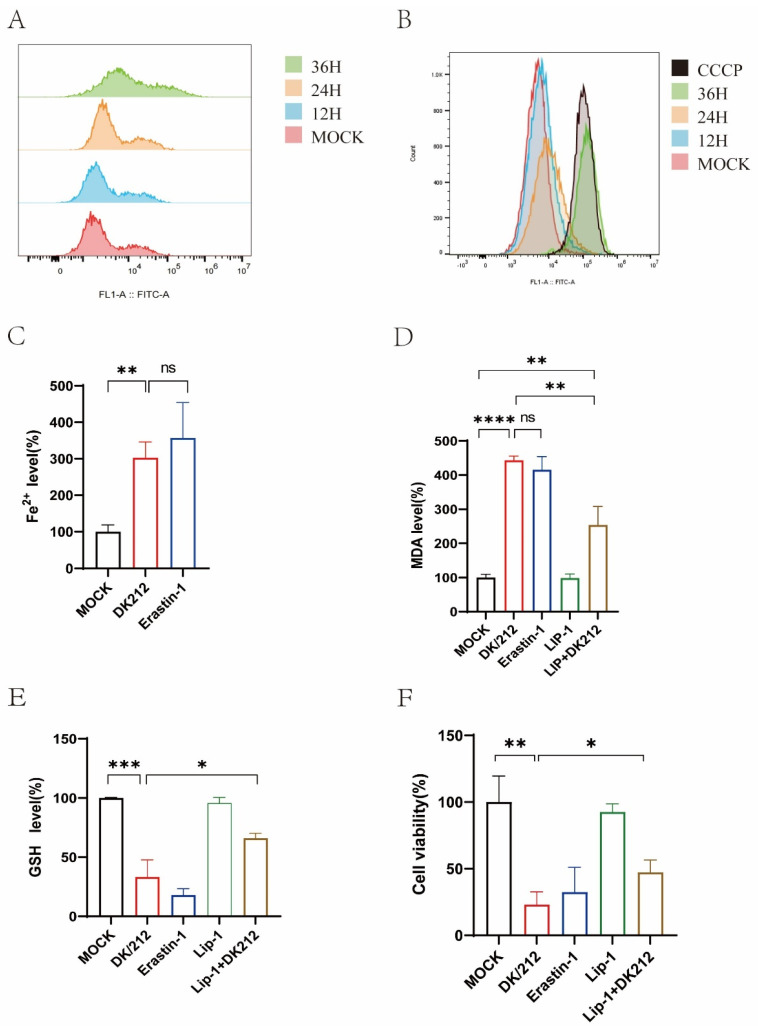
DK/212 AIV infection induced oxidative stress and ferroptosis in A549 cells. (**A**) The ROS level was detected by DCFDA at 12, 24, and 36 h. (**B**) MMP were detected using flow cytometry by JC-1 staining, CCCP as a positive control. (**C**) The level of ferrous irons in cell lysate was detected at 24 h post infection, A549 cells treated by 5 μM Erastin as a positive control. (**D**) The level of MDA was detected at 24 h post infection. A549 cells were treated by 5 μM Erastin or 20 nM liproxstatin-1. (**E**) The level of GSH was detected at 24 h post infection. A549 cells were treated by 20 nM liproxstatin-1 after infection. (**F**) The cell viability was detected at 24 h post infection. A549 cells were treated by 5 μM Erastin or 20 nM liproxstatin-1. All data were expressed as means ± SD; the Student’s *t*-test was used to analyze the differences (* *p* < 0.05, ** *p* < 0.01, *** *p* < 0.001, **** *p* < 0.0001, ns *p* > 0.05).

**Figure 2 ijms-25-03315-f002:**
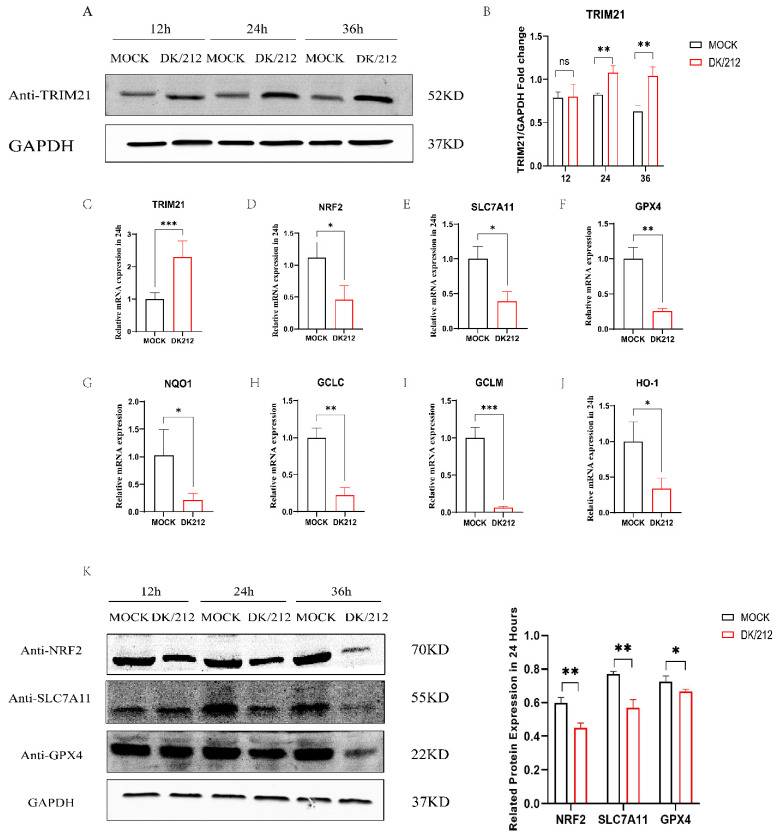
DK/212 AIV infection increased the mRNA and protein level of TRIM21, and reduced mRNA and protein level of related antioxidant genes. (**A**,**B**) A549 cells were infected with DK/212 at an MOI of 0.1. The mRNA and protein level of TRIM21 was detected at 12, 24, and 36 h (**C**–**J**) A549 cells were infected with DK/212 at an MOI of 0.1. The mRNA level of NRF2, SLC7A11, GPX4, NQO1, HO-1, GCLC, and GCLM genes was detected at 24 h. (**K**) A549 cells were infected with DK/212 at an MOI of 0.1. The protein level of NRF2, SLC7A11, and GPX4 was detected at 12, 24, and 36 h. All data were expressed as means ± SD; the Student’s *t*-test was used to analyze the differences (* *p* < 0.05, ** *p* <0.01, *** *p* < 0.001 ns *p* > 0.05).

**Figure 3 ijms-25-03315-f003:**
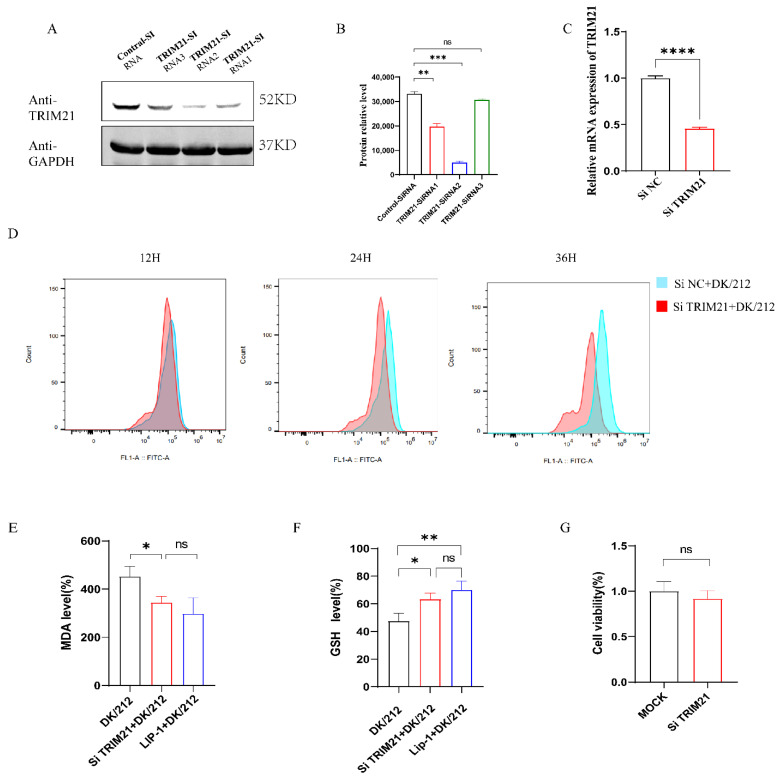
Knockdown of TRIM21 alleviated oxidative stress and ferroptosis induced by DK/212 AIV infection. (**A**–**C**) A549 cells were transfected with TRIM21 si RNA1, si RNA2, and si RNA3. The mRNA and protein level of TRIM21 was detected. (**D**) A549 cells were transfected with TRIM21 si RNA or si NC. The level of ROS was detected at 12, 24, and 36 h after virus infection. (**E**) A549 cells were transfected with TRIM21 si RNA or si NC. The level of MDA was detected at 24 h after virus infection. (**F**) A549 cells were transfected with TRIM21 si RNA or si NC. The level of GSH was detected at 24 h after virus infection. (**G**) The cell viability was detected by cck8 after knockdown TRIM21. All data were expressed as means ± SD; the Student’s *t*-test was used to analyze the differences (* *p* < 0.05, ** *p* < 0.01, *** *p* < 0.001, **** *p* < 0.0001, ns *p* > 0.05).

**Figure 4 ijms-25-03315-f004:**
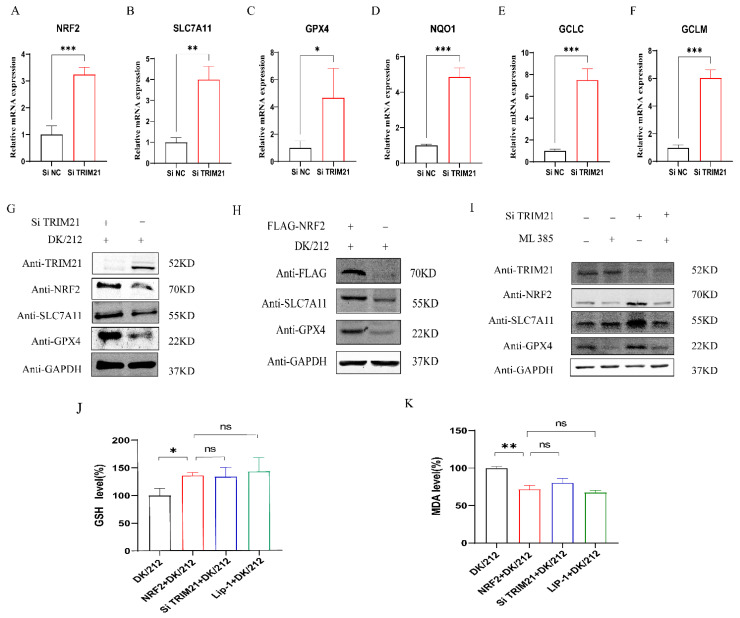
TRIM21 inhibited antioxidant genes and promoted ferroptosis by down-regulating NRF2. (**A**–**F**) A549 cells were transfected with TRIM21 Si RNA. The mRNA level of antioxidant genes was detected at 24 h after DK/212 infection. (**G**) A549 cells were transfected with TRIM21 Si RNA. The protein level of NRF2, SLC7A11, and GPX4 was detected at 24 h after DK/212 infection. (**H**) A549 cells were transfected with FLAG-NRF2 plasmid. The protein level of SLC7A11 and GPX4 was detected at 24 h after DK/212 infection. (**I**) A549 cells were transfected with TRIM21 Si RNA. A549 cells were treated with 2 μM ML385 at 12 h after DK/212 infection. The protein level was detected at 24 h after DK/212 infection. (**J**,**K**) A549 cells were transfected with FLAG-NRF2 plasmid. The GSH and MDA level was detected at 24 h after DK/212 infection. All data were expressed as means ± SD; the Student’s *t*-test was used to analyze the differences (* *p* < 0.05, ** *p* < 0.01, *** *p* < 0.001, ns *p* > 0.05.

**Figure 5 ijms-25-03315-f005:**
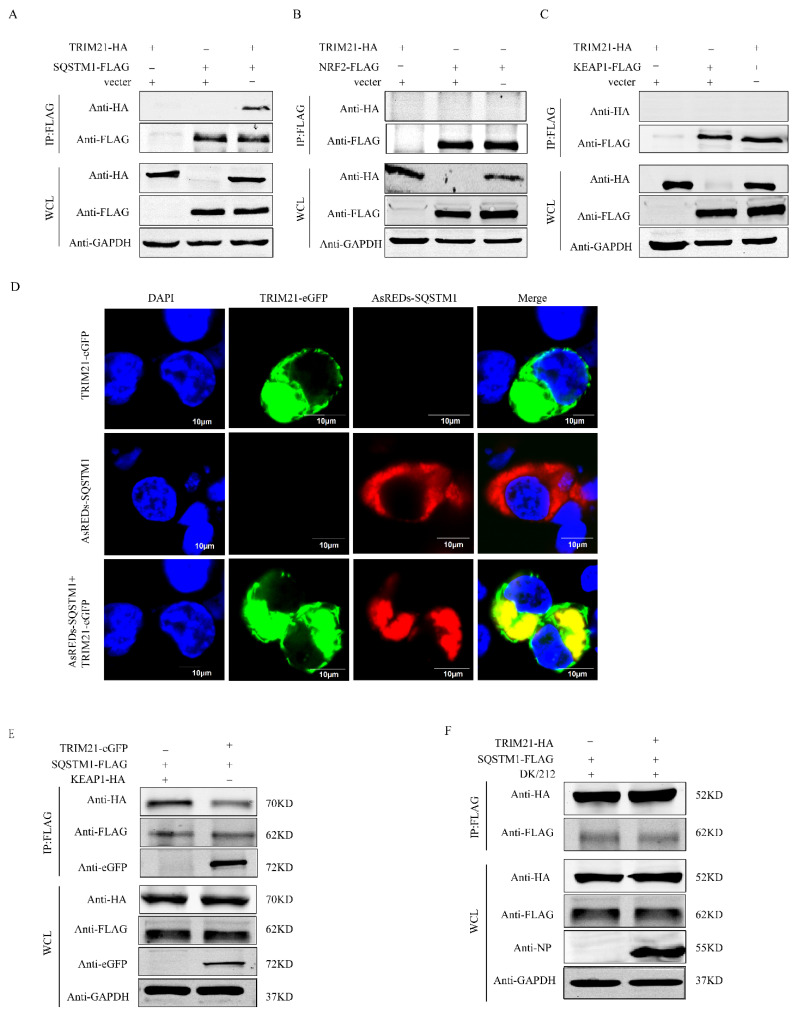
TRIM21 and KEAP1 competed to bind to SQSTM1. (**A**) TRIM21-HA and SQSTM1-FLAG plasmids were co-transfected in 293T cells for 24 h. (**B**) TRIM21-HA and NRF2-FLAG plasmids were co-transfected in 293T cells for 24 h. (**C**) TRIM21-HA and KEAP1-FLAG plasmids were co-transfected in 293T cells for 24 h. (**D**) TRIM21-eGFP and SQSTM1-AsREDs plasmids were co-transfected in 293T cells for 24 h. (**E**) TRIM21-eGFP, SQSTM1-FLAG, and KEAP1-HA plasmids were co-transfected in 293T cells for 24 h. (**F**) SQSTM1-FLAG and TRIM21-HA plasmids were transfected in A549 cells. The protein level was detected at 24 h after DK/212 infection.

**Figure 6 ijms-25-03315-f006:**
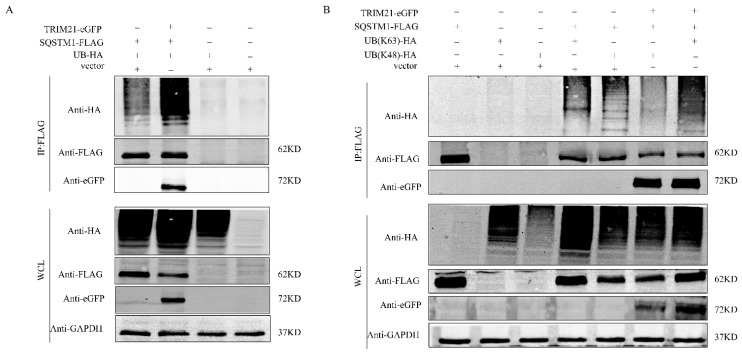
TRIM21 enhanced K63-linked polyubiquitination of SQSTM1. (**A**) TRIM21-eGFP, SQSTM1-FLAG, and UB-HA plasmids were co-transfected in 293T cells. (**B**) TRIM21-eGFP, SQSTM1-FLAG, UB (K48)-HA, and UB (K63)-HA plasmids were co-transfected in 293T cells for detection.

**Figure 7 ijms-25-03315-f007:**
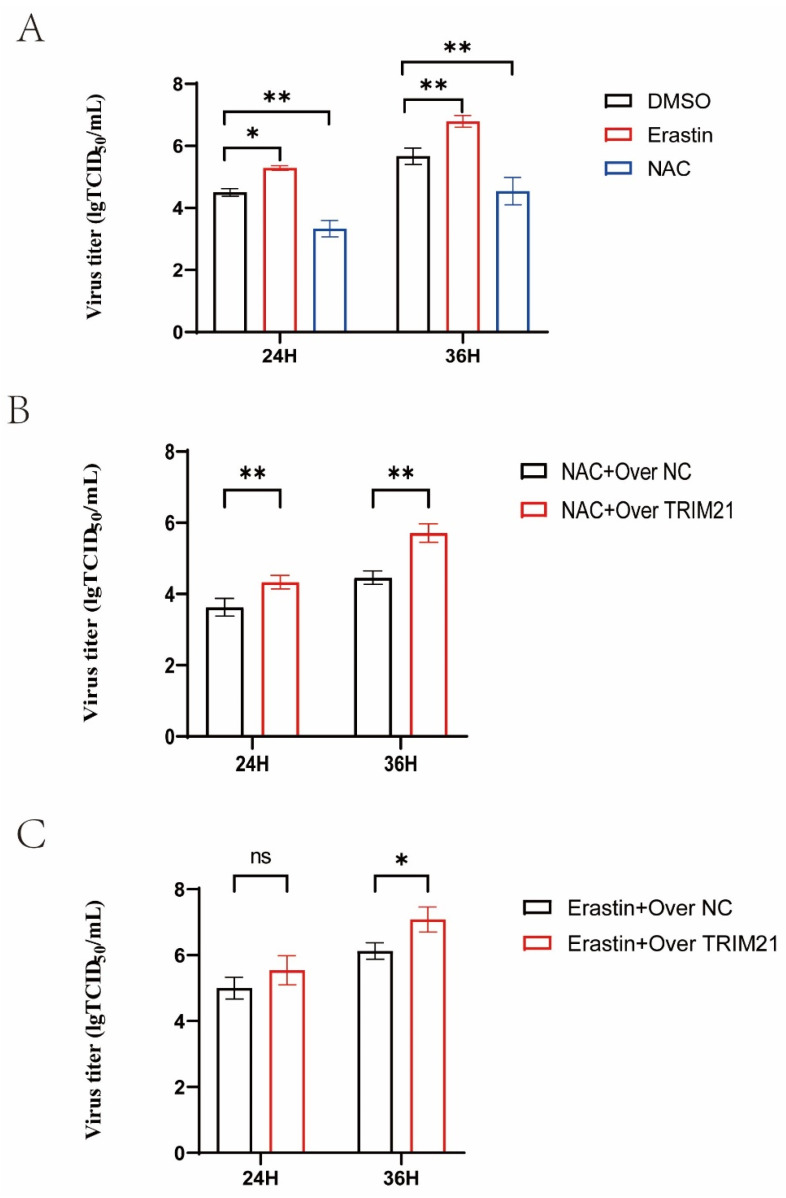
A549 cells were infected with DK/212 at an MOI of 0.1. (**A**) A549 cells were treated with DMSO, or Erastin treated at 12 h after DK/212 infection. (**B**,**C**) A549 cells were transfected with TRIM21 plasmid or NC plasmid. A549 cells were treated with NAC or Erastin at 12 h after DK212 infection. (* *p* < 0.05, ** *p* < 0.01, ns *p* > 0.05).

**Table 1 ijms-25-03315-t001:** Real-time quantitative PCR primers used in this study.

Gene	Forward Primer (5′-3′)	Reverse Primer (5′-3′)
GAPDH	GAAGGTGAAGGTCGGAGTCAAC	CAGAGTTAAAAGCAGCCCTGGT
GCLC	ACAAGAAATATCCGACATAGGAG	ACAAGAAATATCCGACATAGGAG
GCLM	GTTGGAACAGCTGTATCAGTG	CAGTCAAATCTGGTGGCAT
SLC7A11	TTTCTCATTAGCAGTTCCGAT	AGACGCAACATAGAATAACCTG
HO-1	TGCTCAACATCCAGCTCTTTGA	GCAGAATCTTGCACTTTGTTGC
NQO1	GAAGAGCACTGATCGTACTGGC	GGATACTGAAAGTTCGCAGGG
GPX4	GGTAGATTTCAATACGTTCCGGG	TGACAGTTCTCCTGATGTCCAAA
NRF2	AACTCAGCACCTTATATCTCG	GAACAAGGAAAACATTGCCAT
TRIM21	TCAGCAGCACGCTTGACAAT	GGCCACACTCGATGCTCAC

**Table 2 ijms-25-03315-t002:** siRNA used in this study.

siRNA	Primer
siTRIM21-1-F	AGUUAUCCUAUGGUCCUGGGUTT
siTRIM21-1-R	ACCCAGGACCAUAGGAUAACUTT
siTRIM21-2-F	UGGCAUGGUCUCCUUCUACAATT
siTRIM21-2-R	UUGUAGAAGGAGACCAUGCCATT
siTRIM21-3-F	GAGUUGGCUGAGAAGUUGGAATT
siTRIM21-3-R	UUCCAACUUCUCAGCCAACUCTT

## Data Availability

The datasets presented in this study can be found in online repositories.

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
