# Peer review of "TRIM21 Promotes Oxidative Stress and Ferroptosis through the SQSTM1-NRF2-KEAP1 Axis to Increase the Titers of H5N1 Highly Pathogenic Avian Influenza Virus"

_ijms, 2024, doi:10.3390/ijms25063315_

Round 1

Reviewer 1 Report

Comments and Suggestions for Authors

TRIM21, a protein involved in signal transduction and antiviral responses via ubiquitination, has been linked to the disruption of cellular homeostasis during viral infections. Studies on H5N1 highly pathogenic avian influenza virus (HPAIV) infection in A549 cells revealed an increase in TRIM21 expression. Overall, this manuscript provide insight into the role of TRIM21 in oxidative stress and ferroptosis triggered by viral infections, suggesting potential therapeutic strategies for combatting viral diseases.

Some comments:

1.There are lots of abbreviations throughout the manuscript. Before using the abbreviations, the authors need to mention the full form first and then use the abbreviations later. For example, CCCP.

2.Fig 1B: typos: CCCP and not CCP

3.Fig 1D -The MDA level is figure 1E and not D.The authors should swap fig 1D and 1E to match the text in the result section. Hence the figure legend needs to change as well.

4.Fig 2K: The authors mention that the mRNA levels of NRF2, SLC7A11 and GPX4 expression levels were decreased but the protein levels were upregulated after viral infection (line 110-112). Looks like in figure 2K all the proteins levels (NRF2, SLC7A11 and GPX4) were decreased as should be the case after infection. The authors need to clarify the text. 

5.Line 134: MDA and not MAD

6.Figure 3 legends: The legends do not match the corresponding figure. For example, Fig 3G is cell viability but the G in figure legends mentions figure 3D.  Line 143: Figure 3 legends MDA and not MD. Figure legend is mixed up. The authors need to correct the figure legends.

7.Figure 4 legends: The legends do not match the corresponding figure. A should be A-F, B should be G, C should be H and so forth.

8.Line: 183: The authors need to put a reference of the previous study.

9.Figure 5B: The authors mention there is TRIM21-HA co-transfection as shown in figure 5A and 5C but in figure 5B shows no co-transfection with TRIM21-HA and SQSTM1. Please clarify fig 5B with text in the result section.

10.Have the authors performed any experiment with different dose of MOI? Would the change in MOI alter the outcome? 

11. The authors describe they used two-way ANOVA to compare multiple groups in the methods section. However, none of the figure legends state two-way ANOVA as a statistical test, the authors mostly state student’ t test. Can the authors clarify where they have used 2-way ANOVA?

Reviewer 2 Report

Comments and Suggestions for Authors

Wei et al. demonstrated the role of the role of TRIM21 in modulating H5N1 HPAIV-induced oxidative stress and ferroptosis by various assays. The study helps the understanding of the role of TRIM21 in oxidative stress and ferroptosis induced by viral infection. The topic is interesting and important. The manuscript was well designed and written. The quality of the manuscript will be improved if the following concerns can be addressed.

Major concern:

1.       Figure 7B and 7C, cells were infected with DK/212 and cells were transfected with TRIM21 or NC plasmid and the virus titers were tested. To justify the conclusion, the authors may need to transfection with multiple plasmid concentrations and then analyze how these increasing concentrations affect the virus titer.

Minor concerns:

2.       In figure 3A-B, the authors showed the effect of three designed primers. The Si RNAs are in different orders, and it may cause confusion to orders. For consistence, please use Si RNA not SI RNA in figure 3A.

3.       Not all the methods are adequately described. Some methods are needed in the material and method section. please add the transfection and flow cytometry to this section for readers awareness.

Round 2

Reviewer 2 Report

Comments and Suggestions for Authors

this is a revised version and the manuscript is improved. the authors have considered the suggestions and I did not detect concerns in this updated version. Congratulations!